# Endothelial Autophagy Dysregulation in Diabetes

**DOI:** 10.3390/cells12060947

**Published:** 2023-03-21

**Authors:** Yann Salemkour, Olivia Lenoir

**Affiliations:** PARCC, Inserm, Université Paris Cité, 75015 Paris, France

**Keywords:** autophagy, endothelial cells, diabetes

## Abstract

Diabetes mellitus is a major public health issue that affected 537 million people worldwide in 2021, a number that is only expected to increase in the upcoming decade. Diabetes is a systemic metabolic disease with devastating macro- and microvascular complications. Endothelial dysfunction is a key determinant in the pathogenesis of diabetes. Dysfunctional endothelium leads to vasoconstriction by decreased nitric oxide bioavailability and increased expression of vasoconstrictor factors, vascular inflammation through the production of pro-inflammatory cytokines, a loss of microvascular density leading to low organ perfusion, procoagulopathy, and/or arterial stiffening. Autophagy, a lysosomal recycling process, appears to play an important role in endothelial cells, ensuring endothelial homeostasis and functions. Previous reports have provided evidence of autophagic flux impairment in patients with type I or type II diabetes. In this review, we report evidence of endothelial autophagy dysfunction during diabetes. We discuss the mechanisms driving endothelial autophagic flux impairment and summarize therapeutic strategies targeting autophagy in diabetes.

## 1. Introduction

In 2021, diabetes was responsible for 6.7 million deaths worldwide, and current models predict that, by 2045, one-tenth of the world’s population will live with diabetes [1]. Although diabetes is a metabolic disease, the complications of diabetes are essentially macro-vascular complications (cardiomyopathy [2] and arteriopathy [3]) and microvascular complications such as nephropathy [4], retinopathy [5], or neuropathy [6], having devastating consequences on the quality of life of diabetic patients and limiting their life expectancies. Endothelial dysfunction is a key determinant in the pathogenesis of diabetic vascular complications, which justifies the focus of diabetes therapies on the prevention of endothelial damage in diabetic patients. Recent evidence has supported the role of autophagy impairment in endothelial dysfunction in diabetes.

In this review, we summarize the arguments in favor of an essential role for autophagy in the maintenance of endothelial homeostasis and endothelial functions. We then review the evidence for endothelial autophagy dysfunction in diabetes. Finally, we discuss current diabetes therapies that may exert beneficial effects on vessels by preventing impairment of endothelial autophagy. Because most of the data for this review were obtained using in vitro experiments in cells that were a priori competent for insulin signaling and via animal models of non-insulin-dependent diabetes (except when mentioned), and because the pathophysiology of insulin-dependent and non-insulin-dependent diabetes is different, we consider that the findings described in the present review should apply to non-insulin-dependent diabetes.

## 2. Overview of Autophagy

Autophagy was discovered in 1963 by Duve [7] but its functions were only understood in the 1990s, based on yeast studies [8]. It is a highly conserved survival pathway in eukaryotes that acts primarily as an adaptative response to stress and starvation but is also responsible for the recycling of long-lived proteins and misfolded organelles. The word “autophagy” groups together a set of lysosomal degradation processes that can be divided into three main forms: chaperone-mediated autophagy [9], micro-autophagy [10], and macro-autophagy. The essential distinction among the different forms of autophagy is whether or not they are selective. The selective chaperone-mediated autophagy is dependent on heat shock protein 8A (HSP8A, also called HSC70) [11], a chaperone protein that is associated with co-chaperone proteins such as heat shock protein 40 [12] or the carboxyl-terminus of HSC70-interacting protein [13], which recognizes a KFERQ motif on cargo proteins, allowing their lysosomal degradation after association with lysosomal-associated membrane protein 2A and lysosomal HSP8A. Micro-autophagy involves the direct engulfment of cargo proteins by late endosomes or lysosomes after the invagination of their membranes [14]. Macro-autophagy is the best-characterized pathway. It is commonly referred to as autophagy (and it will sometimes be referred to as autophagy hereinafter). It requires an initiation signal that prepares a double-membraned cup-shaped phagophore, which elongates to form a vesicle called autophagosome during the membrane nucleation step that engulfs autophagic substrates; finally, autophagosomes fuse with lysosomes for the degradation of the cargo. Most of the autophagy machinery is recycled during the process.

As the mechanisms of macro-autophagy have been extensively described elsewhere [15,16,17], we will only briefly recount the key proteins involved in the process. Autophagy initiation requires the assembly and activation of the unc-51-like kinase 1 (ULK1) complex. ULK1 activity is regulated by AMP-activated protein kinase (AMPK)-mediated activating phosphorylations at serines 317 and 777 and a mammalian target of rapamycin complex 1 (mTORC1)-mediated inhibitory phosphorylation at serine 757 [18]. ULK1 forms a complex with autophagy-related gene (ATG) 13, a focal adhesion kinase family interacting protein of 200kDa [19] and ATG101 [20]. The activated ULK1 complex translocates to the phagophore initiation site at the endoplasmic reticulum (ER)-mitochondria contact site [21], where it phosphorylates and activates the class III phosphoinositide-3-kinase (PI3K) complex [22]. Class III phosphoinositide-3-kinase vacuolar protein sorting 34 (VPS34) associates with Beclin-1 [23], VPS15, and ATG14L [24] to form the PI3K III complex, which produces phosphatidylinositol-3-phosphate (PtdIns3P) from PtdIns and is crucial for phagophore elongation. ATG9 also appears as a key determinant for phagophore formation [25,26,27,28]. The ATG12–ATG5–ATG16L1 complex is localized to the site of autophagosome formation and, under the influence of PtdIns3P produced by the PI3K III complex, acts as an E3 enzyme to stimulate the conjugation reaction of microtubule-associated protein light chain 3 (LC3) proteins to phosphatidylethanolamine [29,30] (PE) in autophagosome intermediates [31,32,33]. LC3-PE conjugates promote the expansion of the isolation membrane. This reaction is tightly regulated by ATG4 [34], ATG3 [35], and ATG7 [36]. LC3-PE also addresses cargo protein to the emerging autophagosome by direct binding to Sequestosome 1 (P62/SQSTM1) ubiquitin ligase [37]. Recently, Kageyama et al. proposed that p62/SQSTM1 form droplet gels that act as a platform for autophagosome formation [38]. Newly formed autophagosomes, influenced by ATG14 [39] and the SNARE system [40,41], fuse with lysosomes, allowing cargo degradation by its acidic pH.

Autophagy is a dynamic process and monitoring of the autophagic flux in cells may be challenging (see the guidelines for the use and interpretation of assays for monitoring autophagy [42]). Western blot analysis of the expression of the autophagosome form of LC3 (called LC3 II) and other autophagosome proteins such as Beclin1 is commonly used in combination with chemical modulators of autophagy. Under physiological conditions, a low level of autophagy plays a major role in cytoplasm quality control and stress-adaptation to ensure cellular homeostasis. This is particularly true for quiescent and long-lived cells, such as endothelial cells (EC).

## 3. Autophagy in EC

EC line vessels form the first interface between blood components (nutrients, oxygen, signaling molecules, and inflammatory cells) and tissues. Maintaining EC integrity is, therefore, essential for vascular homeostasis. Defects in EC homeostasis or functions, caused by pathological insults or aging, is causative of a variety of diseases including atherosclerosis, metabolic diseases, inflammation, and cancer. Although the role of autophagy in ECs has not been completely elucidated, accumulating evidence has demonstrated that EC-constitutive autophagy is cytoprotective and regulates its function in response to blood flow and metabolic stress (Figure 1). Here, we focus on autophagy functions in ECs that have been demonstrated or confirmed via animal models of the genetic modulation of autophagy.

### 3.1. Autophagy Regulates van Willebrand Factor Maturation and Thrombosis

Weibel–Palade bodies are unique EC organelles whose roles are crucial for blood vessel homeostasis, coagulation, vascular tone, and inflammatory response [43]. The biological responses induced by WPB exocytosis are mediated in part by the von Willebrand factor (VWF) protein [44]. Autophagy regulates VWF maturation and secretion with physiological consequences. In an earlier study, endothelial-selective ATG7-deficient mice exhibited impaired epinephrine-induced VWF release and lowered levels of high molecular weight VWF multimers, with an increased bleeding time as a consequence [45]. Of note, this seminal study [45] was the first to demonstrate a role of autophagy in endothelial cells by using an endothelial-specific genetic loss-of-function approach in vivo. VWF secretion might be mediated, at least partially, by the endothelial soluble NSF attachment protein Receptor (SNARE) protein SNAP23 [46]. Similarly, endothelial-selective ATG5 deficiency leads to increased bleeding time in mice [47,48].

Given the effect of autophagy on VWF maturation, the role of autophagy on thrombosis has been explored. Following FeCl3 injury, mice with endothelial-ATG7 deficiency exhibited prolonged time to carotid and mesenteric artery occlusion and presented smaller thrombi in laser-injured cremasteric arterioles [49]. On the other hand, in a mouse model of venous thrombosis induced by a flow restriction in the inferior vena cava, endothelial-ATG5 deficiency had no consequence on thrombus size or composition [47]. Discrepancies between these two previous articles may have been due to the thrombus models or genetic deletion efficiency, as different endothelial-selective CRE mice were used. Otherwise, autophagy may have different functions in arterial and venous endothelial cells. Finally, ATG7 and ATG5 may also exert other autophagy-independent endothelial functions.

### 3.2. Shear Stress and Autophagy

Blood flow-induced shear stress affects EC membrane mechanosensory elements, such as integrins, G-coupled receptors, and intercellular junction, and promotes their downstream signaling pathways, which are essential for endothelial cytoskeleton rearrangement, cell alignment, endothelial nitric oxide synthase (eNOS) signaling or inflammatory response [50]. Chronic disease, such as type 2 diabetes, severely impairs the adaptation of resistance arteries to changes in blood flow [51]. In previous studies, autophagy appeared to be tightly regulated by blood flow in EC. Guo et al. showed that steady laminar shear stress promoted autophagy and eNOS expression and inhibited endothelin 1 expression in vitro [52]. Lui et al. further demonstrated that laminar flow induces autophagy through the regulation of oxidative stress and Sirt1 activity, which in turn mediates FoxO nuclear translocation, resulting in ATG5, Beclin1, and LC3A mRNA expression [53].

Not only does shear stress induce autophagy, but autophagy is probably a key pathway in the EC response to shear stress. Shear-stress-induced NO production is prevented when autophagy is disrupted [52,54]. Endothelial autophagy deficiency in mice is associated with a discrete defect in aortic EC alignment along the blood flow and with increased EC apoptosis and senescence in aortic zones of high shear stress [55].

Finally, autophagy is directly linked to flow-mediated vascular dilation. The treatment of spontaneously hypertensive rats with the autophagy inducer trehalose enhanced arterial vasodilatation ex vivo [56]. The loss of endothelial ATG5 in mice affects shear stress-dependent signal transduction, as shown by a reduced flow-mediated dilation in isolated arteries and altered flow-mediated remodeling of mesenteric arteries in vivo [48].

### 3.3. Autophagy-Mediated Angiogenesis

Angiogenesis is an essential biological process for new blood vessel formation. Angiogenesis, which is constitutively inhibited in adults, is triggered by environmental changes such as nutrient deprivation, hypoxia, ischemia, or blood flow variation [57]. Its stimulation involves growth factors such as the vascular endothelial growth factor (VEGF) family, angiopoietins, transforming growth factors (TGF), or the fibroblast growth factor (FGF) family [58]. Autophagy and angiogenesis seem to be intrinsically linked; notably, they are linked with VEGF, the most-described pro-angiogenic factor. Senescence in aged EC is also associated with decreased autophagy and the restoration of autophagy in senescence EC-improved angiogenic function [59,60]. Du et al. demonstrated that autophagy induction enhanced tube formation and migration of bovine aortic endothelial cells (BAECs), whereas autophagy inhibition impaired angiogenesis and VEGF-mediated angiogenesis [61]. Recently, Spengler et al. demonstrated that VEGF induced the phosphorylation of ULK1 at Serine 556 via AMPK activation and initiated autophagy in human umbilical vascular endothelial cells (HUVECs), confirming that autophagy is important for functional angiogenesis [62]. This has also been observed in a model of heat-denatured HUVECs, where autophagy activation induced proliferation, migration, and tube-like structure formation [63]. The role of autophagy as a regulator of neoangiogenesis was also demonstrated in vivo in rodents. Endothelial ATG7 deficiency impaired post-ischemic angiogenesis in a mouse model of femoral artery ligation. This regulation is mediated by STAT1 upregulation, which inhibits the pro-angiogenic transcription factor HIF1α [64]. Beclin 1 knockdown, mediated by siRNA systemic delivery, exacerbated neointimal formation in a rat model of carotid injury mediated by endothelial denudation [65]. Endothelial ATG5 deficiency decreases endothelial mitochondrial respiration and ROS production in vitro and causes anarchic retinal neovascularization after ischemia [66]. Finally, we recently provided evidence that endothelial ATG5 deficiency led to impaired VEGF- and flow-mediated neoangiogenesis ex vivo and in vivo in mice, both in microvascular and macrovascular beds [48].

### 3.4. Endothelial Autophagy and the Regulation of Inflammation and Fibrosis

ECs are major component of the inflammatory response [67,68]. In response to inflammatory stimuli, ECs secrete pro-inflammatory cytokines such as interleukin (IL)-1β, IL-6 or tumor necrosis factor α (TNFα), which potentialize the immune response and might increase endothelial permeability [69,70]. The role of endothelial autophagy on cytokine production is not clear; some studies support a pro-inflammatory role of autophagy, while others support an anti-inflammatory role of autophagy. In thrombin-induced inflammation, ATG7-deficient ECs displayed less nuclear factor κ B (NFκB) activity, resulting in decreased inflammation, as shown by decreased IL-6 and monocyte chemoattractant protein 1 secretion and diminution of adhesion molecules intercellular adhesion molecule 1 and vascular cell adhesion molecule 1 [71]. In brain microvascular ECs, oxygen-glucose deprivation/reoxygenation induced ATG3 expression and activated autophagy. ATG3 knockdown had protective effects in this model through the activation of the PI3K/AKT pathway which, in turn, limited programmed cell death and pro-inflammatory cytokine expression [72].

Gui et al. linked autophagy and endothelial-mesenchymal transition (endMT) in chronic renal graft dysfunction. They observed that ATG16L is induced in glomeruli from kidney transplants, while in chronic graft dysfunction, its expression is reduced. ATG16L knockdown in human glomerular endothelial cells promoted endMT via the induction of NFκB signaling and profibrotic cytokines production, promoting the expression of fibronectin and α smooth muscle actin [73].

A link between shear stress, autophagy, and inflammation may exist, as the low shear-stress-mediated anti-inflammatory properties in ECs are mediated by autophagy induction. Indeed, low shear-stress blocked the TNFα-induced inflammation in ECs in vitro, and when autophagy was blocked with BECLIN1 siRNA or 3-methyl-adenine, the anti-inflammatory effect of low shear stress disappeared [74].

Finally, Reglero-Real et al. recently provided evidence that autophagy controlled the recycling process of junctional molecules in endothelial cells, linking diapedesis and autophagy. Indeed, diapedesis depends on ECs’ ability to favor leukocyte adhesion via the expression of P-selectin and to allow transmigration through the reorganization of their intercellular junctions [75]. Inflammation promoted autophagy within endothelial junctions in microvascular venules and defective endothelial autophagy induced accumulation of PECAM-1 and VE-cadherin at EC junctions, resulting in impaired diapedesis in several models of inflammation [76]. Interestingly, the calcium channel transient receptor potential canonical 6 (TRPC6) was found to regulate leukocyte transendothelial migration through the regulation of trafficking of lateral border recycling compartment membrane, which facilitates transendothelial migration [77]. While the link between autophagy and TRPC6 has not been established in [77], it is interesting to note the tight link between autophagy, Ca^2+^ signaling, and TRPC6 [78,79,80].

### 3.5. Endothelial Autophagy Controls Vascular Lipid Homeostasis

Finally, autophagy is closely related to low-density lipoprotein (LDL) homeostasis in ECs. Indeed, endothelial ATG7 deficiency increased transcytosis of LDL but increased oxLDL levels in HUVECs. In vivo, selective endothelial ATG7 or ATG5 deficiency in mice promoted atherosclerosis lesions [55,81].

## 4. Endothelial Autophagy Impairment in Diabetes

Effects of the diabetic environment on EC autophagy have been essentially explored in vitro. We recapitulate here the main recent findings on the regulation of endothelial autophagy by hyperglycemia, advanced-glycation end products, lipids and reactive oxygen species, as the main mediators of EC dysfunction in diabetes (Figure 2). We will not discuss the effects of insulin signaling on endothelial cell autophagy. The interplay between insulin and autophagy has been described in several organs and cells but it has not been studied specifically in endothelial cells. We could postulate that, similar to the mechanism described in other cells, insulin may also inhibit autophagy in endothelial cells via a mechanism involving mTORC1 activity and ULK1 phosphorylation [82,83,84,85,86]. If the in vitro approach does not allow for consideration of the global effects of the diabetic environment, at least it allows us to appreciate the direct effect of a given molecule on EC.

### 4.1. Direct Effects of Hyperglycemia on EC Autophagy

The direct effects of glucose on EC autophagy have been poorly explored, probably because most of the diabetic-mediated EC dysfunction is not directly attributed to glucose but to its downstream effectors. High glucose concentration induced autophagic flux impairment in cardiac microvascular endothelial cells (CMEC) by increasing the phosphorylation of AKT at threonine 308 and serine 473 and, subsequently, activating mTORC1. The inhibition of mTOR by rapamycin in this diabetes-like model restored autophagy and inhibited CMEC apoptosis [87]. In induced pluripotent stem cells (iPSC)-EC, hyperglycemia-mediated endothelial dysfunction was associated with autophagic flux impairment and mitochondrial fragmentation. Calpain inhibition in this condition restored endothelial dysfunction and autophagic flux, prevented mitochondrial fragmentation, and normalized ROS levels, suggesting that glucose-induced calpain activity participated in autophagy impairment in ECs [88].

Streptozotocin is used in rodents to induce pancreatic β cells’ destruction, which produces an experimental model of type 1 diabetes. This model is classically used to assess the pathological consequences of diabetes and is characterized by extreme hyperglycemia in the absence of insulin supplementation. In streptozotocin-treated mice, specific endothelial autophagy deficiency exacerbated renal endothelial lesions and glomerular injury, further supporting the idea that endothelial autophagy exerts endothelial protection in hyperglycemic conditions [89].

### 4.2. Advanced-Glycation End Products and EC Autophagy

Advanced-glycation end products (AGE) are heterogenous molecules produced after the glycation of lipids, protein, or nucleic acid with glucose or other aldose sugar. The accumulation of AGE contributes to the pathogenesis of diabetes [90]. AGE effects are largely mediated by their interaction with their receptor RAGE, a member of the immunoglobulin superfamily [91]. In diabetes, endothelial RAGE activation potentialized inflammation and vascular activation [92]. In non-ECs, AGE and RAGE activate autophagy [93,94,95], and studies have indicated that AGE may also induce autophagy in ECs. Zhang et al. showed that methylglyoxal (an AGE precursor) treatment in HUVECs induced autophagy, probably as a result of mTORC1 inhibition, while the consequences of such autophagy induction were not addressed in that study [96]. Autophagy induction mediated by AGE was also observed by Tong et al., who demonstrated that AGE induced autophagy in HUVECs by a mechanism involving SIRT6-dependent Kruppel-like factor 4 induction. SIRT6 knockdown in vitro and in STZ-induced diabetic mice partially prevented AGE-mediated autophagy induction in ECs. Here, SIRT6 deficiency had no consequence on the cardiac function of diabetic mice [97]. In a murine model of heart failure, RAGE knockout prevented endothelial-to-mesenchymal transition (endMT) and cardiac fibrosis by the diminution of autophagy. Autophagy inhibition with 3-methyladenine or chloroquine alleviated cardiac fibrosis via the prevention of endMT, suggesting that AGE-induced excessive autophagy promoted endMT and cardiac fibrosis [98]. AGE may not just be an inducer of excessive autophagy; it may also promote impaired autophagy. Indeed, long-term exposure of human aortic endothelial cells (HAECs) with AGE led to HAEC apoptosis, which was associated with an increased, but also impaired, autophagic flux. Autophagy blockade with 3-methyladenine prevented AGE-induced apoptosis. Here, the authors showed that AGE led to FoxO1 overexpression in HAECs, which in turn induced autophagy through ATG7 binding but also inhibited the expression of ATG14, thus blocking autophagosome–lysosome fusion.

Similar impairment of autophagosomes–lysosomes fusion was confirmed in freshly isolated aortic ECs from diabetic patients [99]. Interestingly, autophagy could directly limit AGE effects on ECs by regulating their degradation. Far-infrared irradiation protects HUVECs from AGE-induced apoptosis via promyelocytic leukemia zinc finger protein (PLZF)-mediated induction of PI3K and activation of autophagy. In streptozotocin (STZ)-induced diabetic mice, far-infrared irradiation prevented EC apoptosis in intestinal microvessels and inflammation through autophagy activation. In this model, far-infrared irradiation promoted an autophagy-mediated AGE degradation [100].

### 4.3. Lipids, LDL, and EC Autophagy

In diabetic patients, dyslipidemia plays a major role in EC dysfunction and cardiovascular disease. Dyslipidemia affects the majority of type 2 diabetic patients and is also highly prevalent in type 1 diabetic patients, its management often being suboptimal or even neglected in the latter. Diabetic dyslipidemia is characterized by elevated triglycerides, low high-density lipoprotein, and high levels of small dense LDL [101]. The oxidation of LDL is a major contributor to lipid-mediated endothelial dysfunction and is triggered by oxidative stress, inflammation, and endoplasmic reticulum stress and catalyzed by NADPH oxidase (NOX), lipoxygenase (LOX), mitochondrial ROS, or eNOS uncoupling [102,103]. LDL exerts direct effects on ECs via its receptor LDL-R. In HUVECs, LDL-R and insulin receptors form a complex, and LDL stimulation, such as stimulation by insulin, promotes PI3K/AKT activation and subsequent autophagy inhibition [104]. Whether an LDL-mediated autophagy blockade in ECs influences endothelial dysfunction was not addressed, but one could suggest that the blockade of lipophagy, the degradation process of lipid droplets by the autophagy machinery, would promote endothelial dysfunction. In lymphatic ECs, lipophagy impairment resulted in lipid droplet accumulation, diminution of mitochondrial ATP production, and defects in angiogenesis [105]. It might generate analog regulation in ECs during diabetes, as suggested by Zhang et al., where long exposure to ox-LDL impaired lipophagy in HUVECs [106]. Furthermore, autophagy has been linked to high glucose-mediated LDL transcytosis in ECs. In HUVECs, high glucose suppressed the caveolin-CAVIN-LC3B-mediated autophagic degradation of caveolin 1, then induced the increased formation of caveolin at the cell membrane, thus facilitating LDL transcytosis across ECs. Correlating with such findings, an accumulation of lipids was found in the subendothelial space of umbilical venous walls of pregnant women with gestational diabetes mellitus [107]. Low-grade systemic inflammation, which accompanies diabetes, is linked to autophagy-mediated lipid transcytosis: in cardiac microvascular endothelial cells, TNF-α stimulated palmitic acid transcytosis, which further impaired the insulin-stimulated glucose uptake by cardiomyocytes and promoted insulin resistance. In this process, TNF-α stimulated endothelial autophagy and NFκB signaling resulting in an increased expression of fatty acid transporter protein 4 (FATP4) in ECs and palmitic acid transcytosis [108].

Renal tubular lipid accumulation is observed in the kidneys of diabetic patients and impaired lipophagy was correlated to the tubular injury in diabetic kidney disease. The restoration of lipophagy with AdipoRon treatment alleviated fibrosis in the kidneys of db/db mice [109]. While EC autophagy was not specifically addressed in [109], one can ask whether the stimulation of EC lipophagy was implicated in the beneficial effects of AdipoRon, especially because endothelial autophagy exerted anti-fibrotic action in the kidneys of obese mice. Indeed, Takagaki et al. demonstrated that in non-diabetic obese mice, endothelial-selective ATG5 deficiency favors renal fibrosis through the promotion of endMT. The effects of autophagy deficiency on endMT were independent of the TGFβ pathway but IL-6-dependent: IL-6 neutralizing antibody prevented endMT in ATG5 knock-down human microvascular cells and ameliorated metabolic disorders in endothelial-selective ATG5 deficient mice fed with high fat diet [110].

### 4.4. ROS, Mitochondrial Dysfunction, Autophagy, and Mitophagy in EC in the Diabetic Environment

Endothelial reactive oxygen species (ROS) include superoxide anion (O^2−^), hydrogen peroxide (H_2_O_2_), peroxynitrite (ONOO^−^), nitric oxide (NO), and hydroxyl (OH); all of them are products of physiological metabolic processes [111,112,113]. In ECs, ROS are essential for VEGF signaling and eNOS pathway activation, resulting in the vasodilatation of blood vessels, and are essentially produced by nicotinamide adenine dinucleotide phosphate oxidase in physiological conditions [114,115]. ROS accumulation during diabetes is well-described and its implications in vascular complications are established [116,117,118,119]. The first evidence of ROS accumulation in the endothelium during diabetes was described by Hattori et al., who determined that the accumulation of superoxide anion in the aorta of a diabetic rat was responsible for altered endothelium relaxation [120]. A large amount of evidence has linked ROS production and autophagy regulation in non-ECs (reviewed in [121,122]). In HUVECs, palmitic acid promoted ROS accumulation and apoptosis, which was prevented by resveratrol-induced autophagy. This stimulation of autophagy by resveratrol was mediated by TFEB induction (a member of the basic helix-loop-helix leucine-zipper family of transcription factors that is a master regulator of lysosomal biogenesis and autophagic flux). Interestingly, autophagy induction by resveratrol also prevented ROS accumulation in palmitic acid-treated HUVECs, highlighting a bilateral regulation of autophagy and ROS in ECs [123]. In glomerular endothelial cells (GECs), diabetic conditions or glucose treatment induced an ROS accumulation that was associated with decreased NO bioavailability, eNOS inhibition, and autophagy induction [124].

Mitochondria homeostasis was found to be at the interplay between ROS and autophagy in ECs in a non-diabetic context. In microvascular brain ECs exposed to high-salt, ROS accumulation promoted mitochondrial uncoupling protein 2 (UCP2) expression and autophagy. In UCP2-silenced ECs, high-salt-induced ROS was unable to induce autophagy and Beclin 1 overexpression restored EC viability in UCP2-silenced ECs exposed to high salt [125]. The aberrant production of mitochondrial ROS blocked the nuclear translocation of TFEB and induced lysosomal dysregulation and autophagy impairment in the corneal endothelium. Mitochondrial ROS quenching with mitoQ restored endothelial autophagy and lysosomal function in this model [126]. Diabetes-induced EC autophagy impairment was recently related directly to aberrant mitochondrial ROS production and decreased NO bioavailability. Zhao et al. observed that HUVEC exposure to oxidized low-density lipoprotein (ox-LDL) or AGE altered ECs’ autophagic flux. An autophagic flux impairment was also observed in aortic ECs in db/db diabetic mice. An AGE-induced autophagic flux blockade led to increased production of mitochondrial ROS, which reduced eNOS activity by disassociating eNOS dimers and mediated endothelial dysfunction in diabetic mice. The inhibition of autophagy with chloroquine of bafilomycin A1 was sufficient to reduce eNOS dimerization in HUVECs and attenuate acetylcholine-dependent relaxation in the aortas. The stimulation of autophagy by mTOR inhibitor rapamycin or with TFEB overexpression prevented the AGE/oxLDL-mediated accumulation of mitochondrial ROS, increased eNOS dimerization, and attenuated diabetic endothelial dysfunction [127].

### 4.5. EC Mitophagy and Anti-Oxidant in Diabetes

Mitophagy is the selective elimination of damaged mitochondria via autophagy and is essential for mitochondria homeostasis and oxidative stress control [128]. Mitophagy involves phosphatase and tensin homologue (PTEN)-induced putative kinase 1 (PINK1) and PARKIN. When the mitochondria membrane potential decreases, cytosolic PINK1 is stabilized at the mitochondrial surface and forms a complex with translocase of the outer membrane (TOM) proteins. After dimerization, autophosphorylation, and auto-ubiquitinylation, PARKIN is recruited by PINK1 and phosphorylated on its ubiquitin-like sequences resulting in its activation. PARKIN act as a ubiquitin-ligase that recruits the autophagy machinery by binding to autophagy adaptors such as SQSTM1 [129,130]. Another regulation of mitophagy involves BNIP3, which has been shown to function at the outer mitochondrial membrane as a molecular adaptor targeting mitochondria at LC3 molecules [131]. The diabetic environment induces mitophagy impairment in non-ECs and the role of stimulating mitophagy in the prevention of diabetes-induced organ injury is well-described [132,133,134,135,136].

In human aortic endothelial cells (HAECs), metabolic stress, induced by a low dose of palmitic acid, activated PINK1/PARKIN mitophagy, but a high dose of palmitic acid inhibited mitophagy, leading to endothelial injury [137]. In rat aortic endothelial cells, hyperglycemia and hyperlipidemia induced mitochondrial ROS production and apoptosis. Under high-glucose and high-palmitate treatments, the anti-oxidant hydrogen sulfide protected the endothelium by facilitating PARKIN recruitment by PINK1 and, thus, promoting mitophagy [138]. Similarly, AMPK stimulation using Ginseng–Sanqi–Chuanxiong extract, a traditional Chinese medicine, reduced high-glucose- and palmitate-induced mitochondrial ROS accumulation and EC senescence by restoring mitophagy in HAECs [139]. The protective effect of mitophagy induction was highlighted in diabetic retinopathy, where the anti-oxidant notoginsenoside R1 attenuated injury by enhancing PINK1 mitophagy in db/db mice and reducing inflammation and apoptosis in ECs in vitro [140]. BNIP3-mediated mitophagy stimulation using another anti-oxidant molecule, resveratrol, attenuated ox-LDL-induced mitochondrial ROS accumulation and endothelial dysfunction in HUVECs [141]. In brain microvascular endothelial cells, brain-derived neurotrophic factor (BDNF) enhanced BNIP3-mediated mitophagy, limiting BMEC dysfunction under a high-glucose concentration [142].

Limiting the accumulation of mitochondrial ROS in ECs by stimulating mitophagy seems to be a promising approach to limiting diabetes-induced endothelial dysfunction. Anti-oxidants seem to fulfill this role in vitro. If they are well-tolerated in dietary supplements or traditional medicine, and if some of them have an anti-aging role (as has been assumed for years), such anti-oxidants may have a benefit for humans. No such molecule has an approved indication in diabetes. On the other hand, it is very likely that current therapies used to treat diabetic patients modulate autophagy and the endothelial protective effect of these molecules could, therefore, be partly via the modulation of autophagy.

## 5. Current Diabetes Treatment May Prevent Endothelial Autophagy Impairment

The first-line medication for type II diabetes is metformin together with a comprehensive lifestyle. SGLT2 inhibitors and GLP-1 RA are indicated for patients with cardiovascular risk or chronic kidney disease. Here, we discuss evidence that current diabetes treatment may exert vascular protection, at least in part, via the regulation of endothelial autophagy (Figure 3). Other hypoglycemic agents, such as sulfonylureas and thiazolidinediones, should not be considered for patients with cardiovascular risk or chronic kidney disease, as they may be associated with excess cardiovascular events (see the EASD–ADA consensus guidelines 2020–2021) and the links between these drugs and EC autophagy are not discussed herein. In addition, we do not discuss the effects of insulin supplementation on EC autophagy.

### 5.1. Metformin

Metformin is the most commonly used anti-diabetic drug for patients with type II diabetes. It acts as a hypoglycemic actor by increasing glucose uptake and decreasing hepatic gluconeogenesis and insulin resistance. Metformin also participates in endothelial homeostasis regulation [143]. In HUVECs, metformin prevented oxLDL-mediated injuries, such as oxidative stress, apoptosis, or AKT, and eNOS downregulation through SIRT1 overexpression [144]. In a type I diabetes model, metformin enhanced wound healing and angiogenesis through restoration of AMPK and eNOS levels [145]. Metformin has been shown to reduce adverse lipid droplet accumulation and a pro-inflammatory response induced by saturated fatty acid through the restoration of autophagic flux in primary mouse heart endothelial cells [146]. Further study is required to identify if this is due to canonical autophagy or specific lipophagy. The mechanisms by which metformin induces autophagy are linked to AMPK activation [147]. Recently, Ma et al. demonstrated that AMPK activation was dependent on presenilin enhancer 2 (PEN2), a component of the gamma-secretase. A low dose of metformin or glucose starvation induces the binding of PEN2 to ATP6AP1 (subunits of v-ATPase), resulting in the activation of lysosomal AMPK. This reaction was also involved in mTORC1 inhibition [148]. Interestingly, Niu et al. proposed an alternative mechanism, in which the regulation of autophagy by metformin was mediated by the hedgehog (Hh) pathway rather than by AMPK. They demonstrated that, in vitro and in vivo, metformin prevents hyperglycemia-mediated endothelial injuries by downregulating autophagy through Hh-dependent BNIP3 activation [149].

### 5.2. Incretins—GLP-1 RA and DPP-4 Inhibitor

Glucagon-like peptide-1 (GLP-1) and glucose-dependent insulinotropic polypeptide (GIP) are the two main incretins, which are gut hormones that stimulate insulin secretion in a glucose-dependent manner [150]. Because of their effect on glycemic control, those two targets became attractive in type II diabetes treatment. Two forms of action have been developed to target this system—the glucagon-like peptide-1 receptor agonist (GLP-1 RA) that directly activates GLP-1 receptor and the dipeptidyl peptidase-4 (DPP-4) inhibitor that inhibits the enzyme responsible for GIP and GLP-1 inactivation [151]. Both of these inhibitors have demonstrated positive effects in regard to cardiovascular diseases such as hypertension, ischemic heart disease, heart failure, obesity, and diabetes [152,153].

Studies have linked GLP1 signaling and endothelial autophagy, suggesting that GLP1-RA and DPP-4 inhibitors may exert vascular protection through the restoration of a functional endothelial autophagic flux. In HUVECs, GLP-1 stimulated eNOS production and phosphorylation [154] and limited apoptosis and ROS production induced by high-glucose treatment [155]. GLP-1 exerts endothelial protection from oxidant injury by preventing excessive autophagy, which may be dependent on restoring HDAC6 through a GLP-1R-ERK1/2-dependent pathway [156]. Cai et al. demonstrated that GLP-1 reduced oxidative stress and retinal injury, in a model of diabetic retinopathy, by a mechanism that could also involve AKT-ERK1/2-HDAC6-mediated regulation of autophagy [157]. In a high-glucose condition, the DPP-4 inhibitor sitagliptin restored altered autophagy in endothelial progenitor cells (EPCs), which limited apoptosis and ROS production and improved angiogenic function [158]. Liraglutide, a GLP1-RA, limited endothelial mitochondrial stress and excessive PINK1/Parkin-dependent mitophagy in HUVECs treated with high glucose [159]. In a diabetic kidney-disease model, liraglutide preserved endothelial homeostasis by increasing phosphorylated e-NOS. Liraglutide treatment was also associated with a diminution of mTOR activation and an increase in LC3B -II expression, suggesting activation of autophagic flux [160]. Indirectly, the DPP-4 inhibitor alogliptin preserved endothelium-mediated vasodilation in obese mice by stimulating NO production. Zhu et al. demonstrated that this regulation is dependent on autophagy activation in perivascular adipose tissue [161].

### 5.3. SGLT2 Inhibitors

Sodium–glucose cotransporter 2 (SGLT2) inhibitors are the most recent treatments for diabetes. Initially developed for the prevention of diabetic nephropathy, they present beneficial effects on other complications of diabetes and metabolic diseases. Clinical trials reported a protective effect of SGLT2 inhibitors on endothelial function, such as flow-mediated dilation improvement [162,163]. In vitro, SGLT2 inhibitors increased NO bioavailability, prevented inflammation, preserved the endothelial glycocalyx, and promoted angiogenesis in ECs [164], although no direct effect of this treatment on endothelial autophagy has been reported. On the other hand, empagliflozin (an SGLT2 inhibitor) in a non-alcoholic fatty liver disease model reduced steatosis by stimulating autophagy and reducing inflammation and ER stress markers [165]. In the same way, empagliflozin has shown beneficial effects on diabetic glomerulosclerosis via the restoration of glomerular autophagy in type II diabetic mice [166]. Other investigations are required to demonstrate the implication of SLGT2 inhibitors in the regulation of endothelial autophagy and to determine the mechanisms by which this effect is mediated. One may think that SLGT2 inhibitors could restore altered endothelial autophagy by mechanisms involving the energy sensors of the cells, such as AMPK and mTOR.

### 5.4. Anti-Hypertensive Therapies, Angiotensin Receptor Blockers, and Angiotensine Converting Enzyme Inhibitors

Hypertension is a common feature of diabetic patients and is associated with the vascular complications of diabetes. Control of blood pressure is first in line in preventing cardiovascular and microvascular complications in diabetic patients; a large majority of type 2 diabetic patients take hypertensive medication. The angiotensin receptor blockers (ARBs) and the angiotensin-converting enzyme (ACE) inhibitors are able to improve endothelial dysfunction and vascular inflammation in patients with hypertension and other cardiovascular diseases [167]. Furthermore, links between angiotensin signaling and autophagy exist in non-ECs [168,169,170,171]. Notably, the mechanisms of autophagy regulation by the angiotensin system imply mTOR, AMPK [172], and calpains [173].

Whether autophagy is inhibited or activated by angiotensin is still controversial and may depend on the disease context, but the benefits of ARBs have been linked to autophagy modulation in models of cardiac hypertrophy and prostate cancer [174,175]. The question in ECs has only been addressed in a few articles. On the one hand, in HUVECs, angiotensin II could promote autophagy, resulting in decreased NO production, and candesartan inhibited autophagy-mediated altered NO production [172]. In HUVECs and human pulmonary microvascular endothelial cells, angiotensin II also promotes autophagy as a protective mechanism of injury. Autophagy blockade in this context enhanced endothelial cell death [176]. On the other hand, losartan prevented senescence and apoptosis in high-glucose-treated HUVECs through the restoration of impaired autophagy [177].

Renal denervation recently appeared as a promising therapy for type 2 diabetic patients by both reducing blood pressure and improving glucose metabolism, insulin sensitivity and endothelial function in rat models. A link with endothelial autophagy was made by Wang et al., who demonstrated that renal denervation in type 2 diabetic rats increased the expression of angiotensin-converting enzyme 2 (ACE2, whose effects oppose those of angiotensin II) in endothelial cells, which in turn activated AMPK and inhibited mTOR, thus promoting a protective endothelial autophagy [178].

Therefore, vascular protection conferred by blockers of the renin–angiotensin system might be linked to the direct restoration of functional autophagy in ECs.

### 5.5. Statins

Statins are part of the therapeutic arsenal of many diabetic patients. Statins inhibit 3-hydroxy-3-methyl-glutaryl coenzyme-A (HMG-CoA) reductase that results in the decrease of LDL and triglyceride levels [179]. Despite its promising effect on cardiovascular events, prolonged statin treatment has been associated, in a dose-dependent manner, with increased diabetes progression, such as that resulting from insulin resistance or hyperglycemia [180]. The therapeutic roles of statins has partially been attributed to the induction of autophagy by mechanisms implicating the modulation of AMPK and mTOR activity [181]. In HUVECs, atorvastatin stimulated autophagy without affecting apoptosis at a low dose but induced apoptosis and necrosis at higher doses [182]. Simvastatin increased autophagy and lysosome biogenesis by activating TFEB in mouse microvascular endothelial cells (MVECs), protected cells from palmitate-induced NLRP3 inflammasome activation, and altered endothelial permeability [183].

## 6. Concluding Remarks

Patients with diabetes are at heightened risk of adverse microvascular and cardiovascular events. Although cardiovascular-event rates have declined over the past decade, the incidence remains higher for diabetic patients than for nondiabetic patients. Moreover, once a cardiovascular disease develops, the progression of diabetes is exacerbated and outcomes are worsened. Understanding the pathophysiology of diabetes and its vascular complications will foster new treatments to prevent and treat vascular disease in diabetes mellitus. Current therapies that aim to control blood pressure and normalize glycemia and lipidemia reduce the risk of vascular complications. Impaired autophagy has been associated with endothelial dysfunction in diabetes. In this study, we reviewed the most recent data demonstrating that diabetes triggers endothelial autophagy impairment, which participates to diabetes-mediated endothelial dysfunction. Interestingly, most current anti-diabetic drugs exert protection in ECs and these drugs also modulate autophagy. Some in vitro evidence has demonstrated that anti-diabetic drugs could regulate autophagy in altered ECs, suggesting that the vasculo-protective action of anti-diabetic medication may be partly through ECs’ autophagy regulation.

## Figures and Tables

**Figure 1 cells-12-00947-f001:**
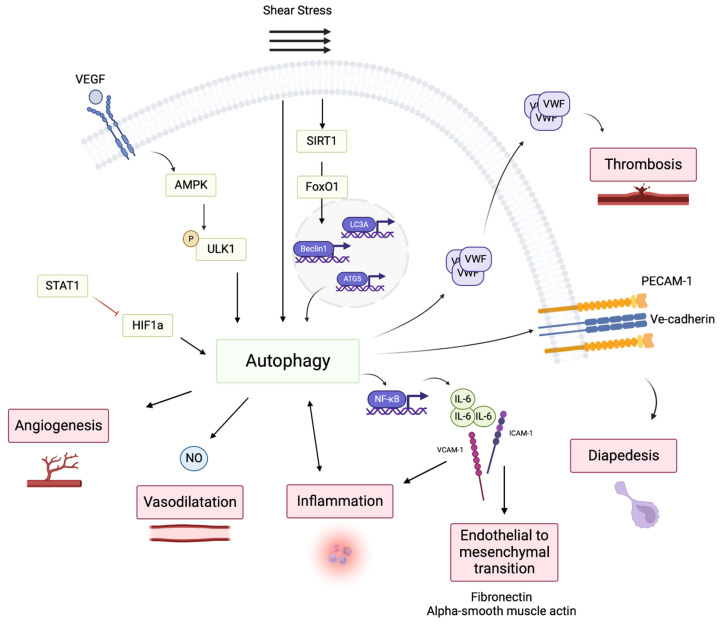
Main regulations of endothelial functions by autophagy. Extracellular signals such as blood flow or angiogenic factors stimulate the autophagic flux in ECs. Autophagy is implicated in the regulation of several EC functions to regulate important vascular functions: angiogenesis, vasodilatation, endothelial-to-mesenchymal transition, inflammation, and diapedesis. (Created with Biorender: https://biorender.com accessed on 10 January 2023).

**Figure 2 cells-12-00947-f002:**
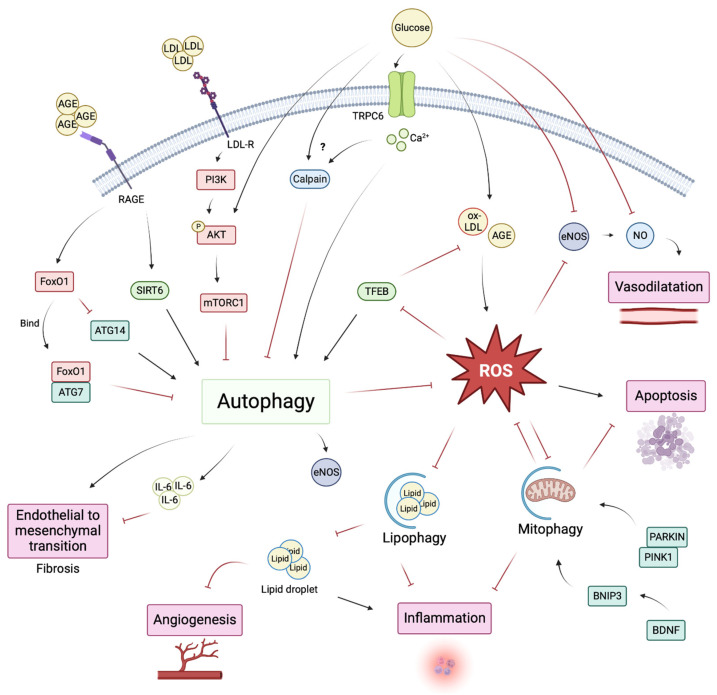
EC autophagy impairment in diabetes. The diabetic environment promote EC dysfunction that is linked to autophagy impairment. (Created with Biorender: https://biorender.com accessed on 10 January 2023).

**Figure 3 cells-12-00947-f003:**
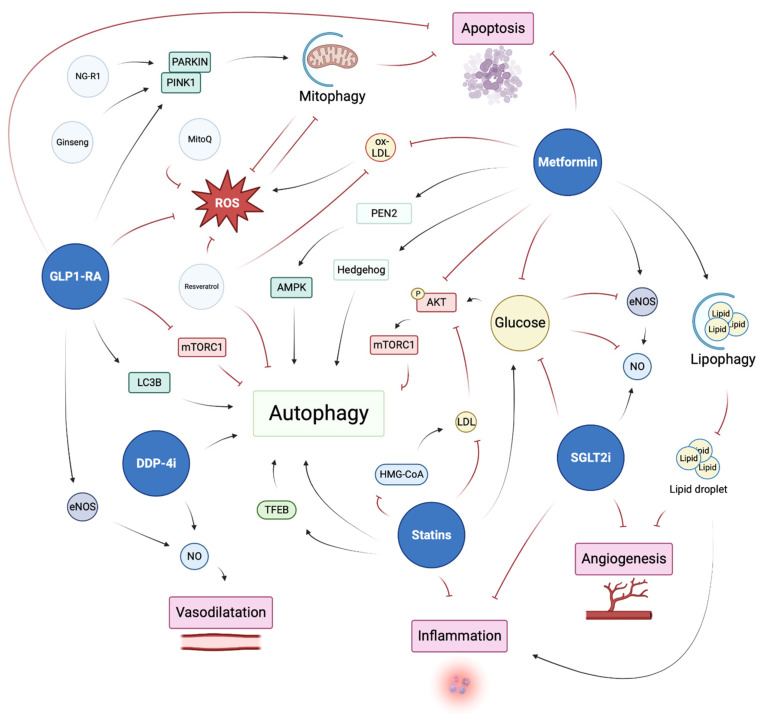
Current diabetes therapies could exert beneficial effects on EC function through modulation of EC autophagy. (Created with Biorender: https://biorender.com accessed on 10 January 2023).

## Data Availability

Not applicable.

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
