# Peer review of "Endothelial Autophagy Dysregulation in Diabetes"

_cells, 2023, doi:10.3390/cells12060947_

Round 1

Reviewer 1 Report

This manuscript is a tentative review the role and mechanisms involved in endothelial cell autophagy dysregulation in diabetes.

Whereas this is a very interesting and important topic which has been neglected in this major metabolic disease, this review is rather superficial and lacks precision in many aspects.

One of the major issues here is that of addressing both IDDM and NIDDM without clearly differentiating between the respective molecular mechanisms which may be involved in autophagy dysregulation in two completely different diseases.

In addition, describing diabetes merely as "the inability of the body to metabolize glucose" is at best completely outdated, but worse, a serious lack of understanding of this disease. It is therefor difficult or even hazardous to try to explain the putative mechanisms involved in autophagy dysregulation...

The schemes presented, although they mention multiple pathways thereby becoming confusing, lack some essential pathways. E.g. in fig. 1 the only extracellular signals mentioned are VEGF, shear stress and vWF... These are obviously not the major ones and certainly not the essential ones affected by diabetes. Hormones and neurotransmitters are lacking.

When addressing the effects of diabetes, besides the confusion between IDDM and NIDDM, in vitro and in vivo are also mixed up. 

In the section devoted to the effects of diabetes treatments, thiazolidinediones are not used anymore for quite a few years because of their severe side-effects and should thus be removed. Other drugs like angiotensin II receptor antagonists (sartans)and ACE inhibitors which have insulin-sensitizing effects and are often associated with other drugs in NIDDM, should be mentioned especially as angiotensin II has been reported to regulate autophagy.

Finally, there are a few errors such as in l. 415 "increasing insulin resistance", l. 343 DDP-4 instead of DPP-4, l. 451 in contradiction with the previous statements...

This manuscript requires serious polishing and major revision before consideration for publication in any good journal

Author Response

We have taken into account the major remarks of the reviewer to improve the quality of the manuscript.

First, we agree that describing diabetes as "the inability of the body to metabolize glucose" is not adequate. This sentence should have been removed during edition of the submitted version and we removed it in the revised version.

The reviewer consider that one of the major issues here is that of addressing both IDDM and NIDDM without clearly differentiating between the respective molecular mechanisms. Autophagy dysregulation in endothelial cells in diabetes condition has only been studied in vitro and in mouse models of diabetes, STZ-induced and db/db mice. None of this model recapitulate IDDM or NIDDM and only address the role of one specific molecule on EC. For instance, STZ-induced mice is considered to be a Type 1 DM but STZ-mice have higher blood glucose than any type 1 diabetic patients and they are not supplemented with insulin which does not recapitulate the effects of insulin treatment. Consequently, we did not differentiate IDDM and NIDDM but instead grouped together articles studying the effects of a given molecule (glucose, AGE, lipid etc...) on EC autophagy. Furthermore, the number of articles describing autophagy dysregulation in EC specifically in diabetes condition are few in number, separating the in vitro and in vivo approaches would have given the impression of fragmentation into too many small parts. Instead, we decided to group together all the data in favor of a mechanism of autophagy dysregulation

The reviewer considers that hormones and neurotransmitters are lacking as regulator of endothelial autophagy. Regulation of autophagy by neurotransmitters and hormones may have been described in other cell types than EC but our bibliography search did not return any article showing regulation of endothelial autophagy by hormones or neurotransmitters in diabetes. Insulin signaling is a well known regulator of autophagy and this have been demonstrated in several cells and organs but not specifically in endothelial cells, which is the specific topic of the review. In case we missed some important article on the regulation of endothelial cell autophagy by hormones or neurotransmitters specifically altered in diabetes condition, could you specify which articles you are referring to, so we can include them in the manuscript? 

Furthermore, in part 3 of the review we chose to focus on autophagy function in EC more than in regulator of EC autophagy and in Figure 1 it is above all because VEGF, shear stress and VWF are regulated by autophagy that they are here.

The reviewer also has concerned about one sentence citing the use of  thiazolidinediones for diabetes treatment. Whereas pioglitazone is not used in France, and TZD  are certainly not the 1st line treatment of diabetes, they still appear in recent guidelines for diabetes treatment. 

We agree that a section on sartans and ACE inhibitors was missing and we added one.

Finally, we made the corrections for the few errors mentioned

Reviewer 2 Report

This review summarises the role of the endothelial autophagy in diabetes. The topic is interesting and well-conducted study and therefore I suggest acceptabling for publication in current form.

Author Response

We thank the reviewer for their comment. We slightly modified the manuscript to address comments of reviewer 1 and 3.

Reviewer 3 Report

The authors have chosen a topic and interesting problem for review. They did a great job in studying literature. The review is a complete and structured presentation of research on this issue. The bibliography is quite large and includes articles from recent years. I would kile to note the high quality and good content of the figures. My comments concern small details:

1. Please check the abbreviations (e.g., WPB (line 107) and NSF (line 112)). There are a lot of abbreviations in the text. I suppose it would make sense to make a list of them.

2. Figures 1 and 2 are located too far from the place of the first mention, while the reference to figure 3 is given after it. I think that Conclusions should not contain any references, please move the reference to Fig. 3 to the main text.

Author Response

We thank the reviewer for their comment. 

We now provide a list of abbreviations for better clarity and moved Figures according to reviewer's comments.

Round 2

Reviewer 1 Report

Dear authors,

Thank you for your comments.

Unfortunately, I am not convinced that they are well-founded. Indeed, IDDM and NIDDM are totally different diseases particularly from an etiopathological and mechanistic point of view. Lack of insulin cannot be confounded with hyperinsulinism which is observed in the initial stage of NIDDM. In IDDM all insulin-mediated pathways are deficient as there is no or almost no insulin secretion. In NIDDM however, whereas most of the metabolic responses to insulin are blunted, this is not the case for its trophic effects. The pathology of these two diseases is therefore clearly different and they can therefore not be discussed as a single disease. Just like bacterial and viral infections...

Clearly, available animal models do not be perfectly mimic type 1 and type 2 diabetes but they allowed significant progress in the understanding of these diseases. Moreover, besides genetic models (db/db mice, Zucker rats,...) there are nutritional models for NIDDM such as e.g. high fat/high glucose fed rats which mimic obesity-driven NIDDM. The lack of data is anyhow  not a valid ground.

Regarding the role of hormones and neurotransmitters, there are several reports. E.g. for angiotensin II (Menikdiwela KR et al. Autophagy in metabolic syndrome: breaking the wheel by targeting the renin-angiotensin system. Cell Death Dis. 2020 Feb 3;11(2):87., Liu D et al. Autophagy contributes to angiotensin II induced dysfunction of HUVECs. Clin Exp Hypertens. 2021 Jul 4;43(5):462-473., ...), neurotransmitters (Wang Y et al. Renal denervation improves vascular endothelial dysfunction by inducing autophagy via AMPK/mTOR signaling activation in a rat model of type 2 diabetes mellitus with insulin resistance. Acta Diabetol. 2020 Oct;57(10):1227-1243. ), estrogen (Meng Q et al. Estrogen prevent atherosclerosis by attenuating endothelial cell pyroptosis via activation of estrogen receptor α-mediated autophagy. J Adv Res. 2020 Aug 24;28:149-164.).

The point is that apart from the factors you indicate in fig. 1 many other potential factors may be involved as well based on their involvement in autophagy in other cell types.

Finally, by restricting the scope of their review, the authors forced themselves into a situation which led to confusion.

Author Response

Dear reviewer,

We appreciate your insight on the physiopathologie of IDDM and NIDDM. While these diseases cannot be regarded as one disease, I want to emphasize that the focus of this review is not on the pathophysiology of diabetes, but on the consequences of the diabetic environment on a particular mechanism of cellular homeostasis, which is autophagy in a specific cell type, namely the endothelium. Although there are distinctions between IDDM and NIDDM, both diseases share common mechanisms of endothelial injury, such as AGEs and lipids, that can result in endothelial dysfunction. In this context, it is important to note that our review aims to discuss endothelial damage in diabetic patients, which occurs in the long term of diabetes. At this stage, all IDDM patients have insulin therapy and, therefore, may have functional insulin signaling.

Regarding the NIDDM models, I acknowledge that some genetic and nutritional models exist and are useful for studying vascular complications of diabetes. However, for the purpose of this review, we do not question the existence of such models but aim to focus on endothelial autophagy in the diabetic environment. It is worth noting that there is no article describing endothelial autophagy that used NIDDM models, except for the ones we already mentioned in the review.

Regarding the regulation of endothelial autophagy by hormones, while the regulation of autophagy by angiotensin is well described, it has not been well documented in endothelial cells, except for a study by Chen et al.  and Liu et al., both of them discussed in the review. Additionally, we have included the reference by Wang et al. on renal denervation in the type 2 diabetes model.

It is important to clarify that the last article cited on autophagy regulation by estrogen in atherosclerosis is not within the scope of this review. Atherosclerosis is not addressed in this review, and there are already existing reviews on endothelial autophagy and atherosclerosis (PMID: 34856194 PMID: 32671753 PMID: 35096837 PMID: 35402552 PMID: 33807637).

Finally, I would like to note that we chose to focus our review on autophagy dysregulation in endothelial cells in the diabetes context to avoid redundancy with existing publications. The role of autophagy in vascular disease has already been reviewed elsewhere (PMID: 25634970 PMID: 35805165 PMID: 33934518 PMID: 33837545 PMID: 30692642 PMID: 28546358). Similarly, autophagy in diabetes complications has also been reviewed elsewhere (PMID: 36791919 PMID: 35988870 PMID: 34572148 PMID: 34646830 PMID: 34234681).

Thank you for your time and attention to this matter.

Sincerely,

Dr Olivia Lenoir

Round 3

Reviewer 1 Report

Dear authors,

Your comments are well taken but does not solve the issue of the difference regarding the pathophysiological between IDDM And NIDDM which is the key issue here. Indeed, one cannot, as I pointed out earlier, discuss a consequent feature of these without taking these differences into account.

If there are not enough data available yet concerning autophagy in endothelial cells in these two diseases, it is obviously too early to write a review ! 

Author Response

Dear reviewer, To clarify the point, we added the following sentence in the introduction:   "Because most of the data were obtained using in vitro experiments in cells a priori competent for insulin signaling and with animal models of non-insulin-dependent diabetes except when mentioned, and because the pathophysiology of insulin-dependent and non-insulin-dependent diabetes is different, we should consider that findings described in the present review should apply to non-insulin dependent diabetes."